# Cardiac Arrest Survival Postresuscitation In-Hospital (CASPRI) Score Predicts Neurological Favorable Survival in Emergency Department Cardiac Arrest

**DOI:** 10.3390/jcm10215131

**Published:** 2021-10-31

**Authors:** Jeffrey Che-Hung Tsai, Jen-Wen Ma, Shih-Chia Liu, Tzu-Chieh Lin, Sung-Yuan Hu

**Affiliations:** 1Department of Emergency Medicine, Taichung Veterans General Hospital, Puli Branch, Nantou 545, Taiwan; erdr2181@gmail.com; 2School of Medicine, National Yang Ming Chiao Tung University, Taipei 112, Taiwan; horsword70@gmail.com (J.-W.M.); h852@vghtc.gov.tw (T.-C.L.); 3School of Medicine, National Chung Hsing University, Taichung 402, Taiwan; 4Department of Emergency Medicine, Taichung Veterans General Hospital, Taichung 407, Taiwan; 5Institute of Medicine, Chung Shan Medical University, Taichung 402, Taiwan; 6School of Medicine, Chung Shan Medical University, Taichung 402, Taiwan; 7Department of Industrial Engineering and Enterprise Information, Tunghai University, Taichung 407, Taiwan; liushihchia@gmail.com; 8College of Fine Arts and Creative Design, Tunghai University, Taichung 40705, Taiwan; 9Department of Nursing, College of Health, National Taichung University of Science and Technology, Taichung 404, Taiwan

**Keywords:** in-hospital cardiac arrest, emergency department, survival, neurological outcome, coronary angiography, coronary reperfusion, targeted temperature management, post-cardiac arrest care

## Abstract

Background: This study was conducted to identify the predictive factors for survival and favorable neurological outcome in patients with emergency department cardiac arrest (EDCA). Methods: ED patients who suffered from in-hospital cardiac arrest (IHCA) from July 2014 to June 2019 were enrolled. The electronic medical records were retrieved and data were extracted according to the IHCA Utstein-style guidelines. Results: The cardiac arrest survival post-resuscitation in-hospital (CASPRI) score was associated with survival, and the CASPRI scores were lower in the survival group. Three components of the CASPRI score were associated with favorable neurological survival, and the CASPRI scores were lower in the favorable neurological survival group of patients who were successfully resuscitated. The independent predictors of survival were presence of hypotension/shock, metabolic illnesses, short resuscitation time, receiving coronary angiography, and TTM. Receiving coronary angiography and low CASPRI score independently predicted favorable neurological survival in resuscitated patients. The performance of a low CASPRI score for predicting favorable neurological survival was fair, with an AUROCC of 0.77. Conclusions: The CASPRI score can be used to predict survival and neurological status of patients with EDCA. Post-cardiac arrest care may be beneficial for IHCA, especially in patients with EDCA.

## 1. Introduction

Patients who have had an in-hospital cardiac arrest (IHCA) have a high mortality rate [1,2,3,4,5], and the etiology and treatment strategy for these patients differ from those of patients with out-of-hospital cardiac arrest (OHCA). Cardiac arrest in the emergency department (ED) comprises about 10–19% of IHCA events [2,5,6] and is linked with a higher chance of shockable initial rhythms and a higher survival rate compared with patients sustaining cardiac arrest in other locations in the hospital [2]. However, ED cardiac arrest (EDCA) is rarely examined as a distinct group [2,5,7], and the risks of survival and neurological outcome are rarely discussed. Knowledge of the risk factors related to survival and neurological outcome may help to optimize the treatment strategy, including prevention, resuscitation efforts, and post-cardiac arrest care in EDCA.

The cardiac arrest survival post-resuscitation in-hospital (CASPRI) score has been used to predict favorable neurological survival [8]. This score has been validated in an Asian population [9], and is recommended as a good tool for estimation of survival to hospital discharge, with a favorable neurological outcome in patients with return of spontaneous circulation after in-hospital cardiac arrest [4]. It is not known whether the CASPRI score can be applied to EDCA for prognosis prediction. We performed this retrospective study to identify the predictive factors of survival and favorable neurological outcome in patients with EDCA. We also evaluated whether the CASPRI scoring system could be used in EDCA.

## 2. Materials and Methods

Taichung Veterans General Hospital (TCVGH) is a medical center located in central Taiwan. It has 1500 beds and an annual ED volume of 66,000 visits. We retrospectively retrieved information from electronic medical records (EMR) of ED patients in TCVGH, who received cardiopulmonary resuscitation (CPR) from 1 July 2014 to 30 June 2019. Patients who visited the ED for trauma, for out-of-hospital cardiac arrest (OHCA), or had signed a “do-not-resuscitate (DNR)” order were excluded. We also excluded patients who were younger than 18 years old, pregnant, a prisoner, had a psychiatric disease, or had human immune deficiency virus (HIV) infection. Patients who were transferred to another hospital after being resuscitated were also excluded. We did not exclude the patients who had been successfully resuscitated from an OHCA event in other hospitals, and were transferred to our ED with spontaneous circulation on arrival. All identifiable patient information was deleted before data analysis.

Two research assistants, both qualified nursing practitioners with at least 5 years of experience in emergency medicine, reviewed the medical records and abstracted the data on a structured data sheet using the Utstein-style elements of in-hospital cardiac arrest [10]. In addition to accessing the EMR of the ED information system, the research assistants also queried the inpatient information system for records related to post-resuscitation care and outcomes. Two board-certified emergency physicians confirmed the quality of the data using established criteria.

The CASPRI predictors were grouped if they had the same CASPRI score, and the CASPRI scores were grouped to avoid small numbers (fewer than 4 cases) in each group. Candidate predictive factors for survival to discharge, as well as favorable neurological outcome (defined as a cerebral performance category score of 1 or 2) were analyzed. Continuous data were expressed as mean ± standard deviation (SD). Categorical data were expressed as number and percentage. Chi-square tests or Fisher’s exact test were used to compare categorical features, and unpaired t test was used to compare continuous features. *p* values < 0.05 were considered statistically significant. Variables with a *p* value of <0.20 in the univariate analysis were entered into a logistic regression model. We used the area under the receiver operating characteristic curve (AUROCC) to evaluate the predictive powers of the CASPRI scoring system. Analyses were performed using the Statistical Package for the Social Science (IBM SPSS version 22.0; International Business Machines Corp, New York, NY, USA).

This study was approved by the institutional review board (IRB) of Taichung Veterans General Hospital (IRB number: SE20226A).

## 3. Results

We collected a total of 322 patients with non-traumatic cardiac arrest in the ED. Male gender was predominant (68.6%), and 16 (5.0%) had had an initial OHCA event, and was referred to the ED after successful resuscitation in other hospital. The pre-arrest cerebral performance category (CPC) scores of 1, 2, 3, and 4 accounted for 33.5%, 34.8%, 21.7%, and 7.8% of the patients, respectively. Most of the cardiac arrests occurred in the resuscitation room (71.7%), followed by the observation room (20.5%), and the imaging room (3.4%). Six (1.9%) cardiac arrest events occurred outside of the ED when patients were transferred to scope rooms or radiology department for diagnostic/interventional procedures. The most common initial rhythms were pulseless electrical activity (68.1%), followed by asystole (18.3%), ventricular fibrillation (6.5%), and pulseless ventricular tachycardia (6.5%). One hundred and sixty-eight patients (52.2%) were successfully resuscitated (defined as recovery of spontaneous circulation (ROSC) for at least 20 min), and in 3 patients the resuscitation effort was stopped due to DNR order before or after this event. Eighty-nine patients (27.6%) survived to discharge, while 36 patients (11.2%) had favorable neurological outcomes (CPC of 1 or 2) at discharge. A total of 44 patients (13.7%) survived for more than one year, and the majority of these patients (32, 72.7%) had favorable neurological outcomes. A flowchart of baseline characteristics and outcomes is illustrated in Figure 1.

Survival of EDCA was associated with presence of preexisting conditions (heart failure, myocardial infarct, hepatic failure, hypotension/shock, metabolic illnesses, diabetes mellitus, sepsis, and renal failure), non-respiratory causes of cardiac arrest, initial rhythm of ventricular fibrillation or pulseless ventricular tachycardia, receiving coronary angiography (urgent or delayed), attempts of coronary reperfusion, and targeted temperature management (TTM). The cardiac arrest survival post-resuscitation in-hospital (CASPRI) score was also associated with survival, and the CASPRI scores were lower in the survival group (mean ± SD, 15.3 ± 6.4 vs. 19.4 ± 5.4, *p* < 0.001) (Table 1).

In patients successfully resuscitated, favorable neurological survival was associated with absence of hypotension/shock, non-respiratory cause of cardiac arrest, receiving coronary angiography, and attempts of coronary reperfusion (Table 2). Three components of CASPRI score (initial rhythm of witnessed or non-witnessed ventricular fibrillation/pulseless ventricular tachycardia, pre-arrest CPC score of 1 or 2, and absence of factors of mechanical ventilation/sepsis/hypotension or hepatic failure/malignancy prior to arrest) were also associated with favorable neurological survival, and the CASPRI scores were lower in the favorable neurological survival group (mean ± SD, 11.1 ± 5.6 vs. 17.8 ± 6.1, *p* < 0.001) (Table 3).

Multivariate logistic regression analysis showed that the independent predictors of survival were presence of hypotension/shock, metabolic illnesses, short resuscitation time, receiving coronary angiography, and TTM. In resuscitated patients, the independent predictors of favorable neurological survival were receiving coronary angiography (urgent vs. other groups: OR, 95% confidence interval of 5.5, 1.8–16.8), and low CASPRI score (0–9 vs. other groups: OR, 95% confidence interval of 9.2, 2.2–37.4, 10–14 vs. other groups: OR, 95% confidence interval of 7.7, 2.1–28.2, respectively). The AUROCC of low CASPRI score for predicting favorable neurological survival was 0.77 (95% confidence interval: 0.68–0.85) (Table 4).

## 4. Discussion

One of the components of in-hospital cardiac arrest that distinguishes it from out-of-hospital arrest is that the former may result from progressively worsening underlying disease, whereas the latter is often sudden and unpredictable [4]. The mixture of characteristics in out-of-hospital and in-hospital arrest populations explains why cardiac arrest in the ED has a higher survival rate compared with arrest in other locations of the hospital. However, it also raises the question as to whether the predictive factors of survival and favorable neurological outcome in patients with IHCA are applicable to EDCA.

The presence of pre-existing co-morbidities has been shown to be associated with poor survival in patients sustaining IHCA [11,12,13], but our study found that presence of hepatic failure, hypotension/shock, metabolic illnesses, diabetes mellitus, sepsis, or renal failure predicted survival after EDCA, and presence of hypotension/shock and metabolic illnesses were also independent predictors of survival. Patients with these medical conditions presenting with cardiac arrest in the ED might be in the early stage of time-sensitive deterioration in their disease process, which allows early recognition and immediate treatment in the ED setting, and leads to a higher chance of survival. Patients without these medical conditions might lack the reversibility by agile response in the ED, and had lower survival rates than their counterparts.

However, our study showed that absence of hypotension/shock predicted favorable neurological outcome. The absence of factors of mechanical ventilation/sepsis/hypotension or hepatic failure/malignancy prior to arrest, which is one of the components of the CASPRI score, was also associated with favorable neurological survival. Only eleven of one hundred and sixty-eight resuscitated patients received TTM in our study. Further research is warranted to clarify the role of this neuroprotective treatment for EDCA in patients with these medical conditions.

IHCA survival improved from 13.7% in 2000 to 26.7% in 2019 [1,3], but decisions pertaining to not attempting resuscitation and termination of cardiopulmonary resuscitation remain a challenge for clinicians and patients’ families [14]. Systems designed to predict survival and neurological outcome, such as the CASPRI score, could be used to optimize medical resources and enhance communication with patients and families. The strength of the CASPRI scoring system is that it focuses on the 10% of patients successfully resuscitated from an in-hospital cardiac arrest who have a >70% probability of favorable neurological survival to discharge [8]. The CASPRI scoring system was developed using the United States’ Get With the Guideline (GWTG) registry. The total score is calculated by summing the scores of eleven variables, including age, initial arrest rhythm or time to defibrillation, pre-arrest cerebral performance category score, hospital location, duration of resuscitation, presence of mechanical ventilation, renal insufficiency, hepatic insufficiency, sepsis, malignancy, and hypotension prior to the arrest. Although the original model of the CASPRI score did not include patients with EDCA, we found that the CASPRI score can be used to predict the survival and neurological status of patients with EDCA. The CASPRI score was associated with survival, and the CASPRI scores were lower in the survival group in all patients. CASPRI scores were lower in the favorable neurological survival group of patients who were successfully resuscitated, and the multivariate logistic regression analysis showed that CASPRI score was an independent predictor of favorable neurological survival in resuscitated patients. The performance of low CASPRI score for predicting favorable neurological survival was fair, with an AUROCC of 0.77. The application of the CASPRI score as a predictive tool for EDCA could provide accurate prognostication based on precise information about the likelihood of survival and neurological outcome, and may provide critical information to facilitate shared decision-making for cardiopulmonary resuscitation [15,16], such as decisions related to choosing an end-of-life plan, or other aggressive resuscitation efforts.

The management in the post–cardiac arrest period of IHCA should focus on the precipitating cause, hemodynamic and respiratory support, and neuroprotective care [4]. However, this concept is generally derived from evidence obtained from studies on OHCA. Coronary angiography is well known to be associated with superior outcomes in survivors of OHCA [17,18,19], but its role in IHCA has rarely been discussed. Our study found that receiving coronary angiography (urgent or delayed), attempts of coronary reperfusion, and TTM were associated with survival, and receiving coronary angiography and TTM were also independent predictors of survival for all patients. Receiving coronary angiography and attempts of coronary reperfusion were associated with favorable neurological survival, and receiving coronary angiography was also an independent predictor of favorable neurological outcome in successfully resuscitated patients. These findings appear to support the benefits of post-cardiac arrest care for IHCA, especially in patients with EDCA.

Targeted temperature management after cardiac arrest remains the primary neuroprotective approach following out-of-hospital cardiac arrest [20], but results regarding use of TTM in IHCA are inconsistent. Chan et al. found that use of TTM was associated with a lower likelihood of survival to hospital discharge and a lower likelihood of favorable neurological survival in patients with IHCA [21], while other researchers found that the beneficial effects of TTM for patients with IHCA were not significantly different from OHCA, especially when baseline factors were matched [22,23,24]. In our study, TTM was favorable for survival in EDCA, but it was not predictive of favorable neurological outcome in resuscitated patients. Although not every individual component of the CASPRI score was associated with favorable neurological survival, a summation score of all components showed a significantly predictive value in our study. Development of a summation score of post-resuscitation processes may be warranted in future research.

Survival from cardiac arrest was higher in EDs than cardiac arrests in intensive care units (ICU-CA), but their risks factors of survival and favorable neurological outcome seem be similar. Treating ICU-CA as unique entity, Roedl and colleagues found that the SOFA score and liver failure after ICU-CA are independent predictors of mortality [25]. Leloup and colleagues found that six-month survival with no or moderate functional sequelae was correlated with a number of organ failures ≤2 when cardiac arrest occurred, resuscitation time ≤5 min, shockable rhythm cardiac arrests, etiology related to the life-sustaining devices in place, absence of preexisting disability or disease deemed fatal within 5 years, and sedation [26]. Our study showed that survival of EDCA was associated with causes of cardiac arrest, shockable rhythm, and short resuscitation time. We also found that post-resuscitation processes (coronary angiography and TTM) were also associated survival, which was not mentioned in researches of ICU-CA.

Generalization of the results of this study might be limited since it was retrospective and information was collected from a single institution. The chart review method used in this study is subject to various potential shortcomings, including inaccuracy and incompleteness in vital sign measurements and the recording of medical events, and inconsistent criteria for ordering certain examinations and identifying abnormalities during these examinations. We attempted to minimize the limitations of the retrospective medical chart review by asking experienced nursing practitioners to retrieve the data from medical information systems, and having board-certified emergency physicians confirm the quality of the data.

## 5. Conclusions

The independent predictors of survival were presence of hypotension/shock, metabolic illnesses, short resuscitation time, receiving coronary angiography, and TTM. The independent predictors of favorable neurological survival in resuscitated patients were receiving coronary angiography and low CASPRI score. The CASPRI score can be used to predict survival and neurological status of patients with EDCA. The performance of a low CASPRI score for predicting favorable neurological survival was fair. Post-cardiac arrest care may be beneficial for IHCA, especially in patients with EDCA.

## Figures and Tables

**Figure 1 jcm-10-05131-f001:**
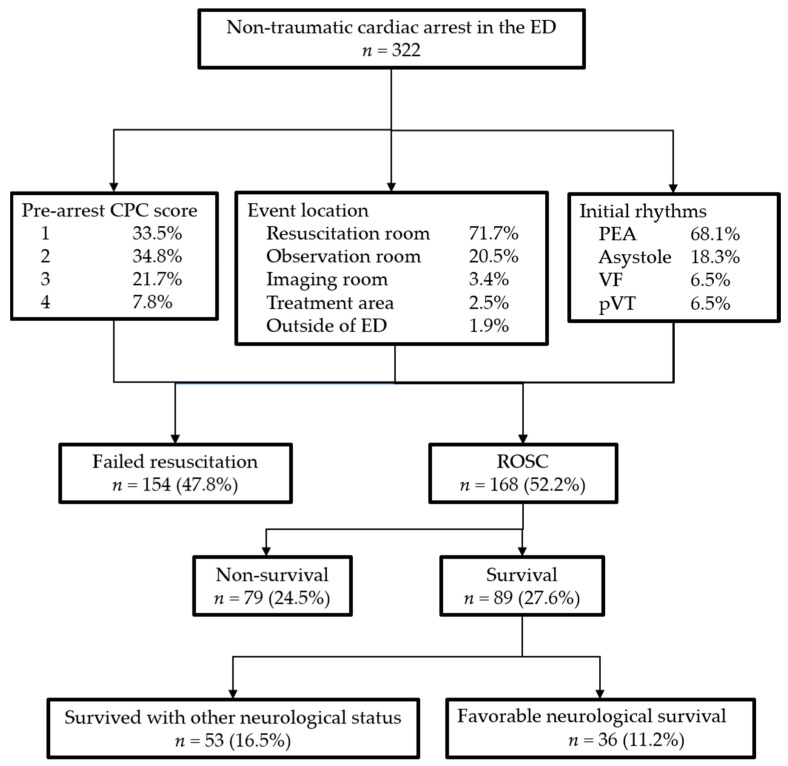
A flowchart of baseline characteristics and outcomes. Abbreviations: CPC: Cerebral performance category; ED: Emergency Department; PEA: pulseless electrical activity; VF: ventricular fibrillation; pVT: pulseless ventricular tachycardia; ROSC: return of spontaneous circulation.

**Table 1 jcm-10-05131-t001:** Clinical characteristics and Utstein style elements of survival and non-survival IHCA patients.

Patient Data	All (*n* = 321)	Survival (*n* = 89)	Non-Survival (*n* = 232)	*p* Value
Age, mean (S.D.)	67.2 (15.5)	64.9 (15.6)	68.1 (15.8)	N.S.
Sex				N.S.
Male (n, %)	221 (68.8%)	57 (64.0%)	164 (70.7%)	
Female	100 (31.2%)	32 (36.0%)	68 (29.3%)	
OHCA				0.155 *
Yes	16 (5.0%)	7 (8.0%)	9 (3.9%)	
No	303 (95.0%)	81 (92.0%)	222 (96.1%)	
CPC before cardiac arrest				N.S.
1	107 (34.1%)	34 (38.6%)	73 (32.3%)	
2	112 (35.7%)	30 (34.1%)	82 (36.3%)	
3	70 (22.3%)	20 (22.7%)	50 (22.1%)	
4	25 (8.0%)	4 (4.5%)	21 (9.3%)	
Preexisting conditions				
Heart failure				0.005
Yes	37 (11.6%)	18 (20.2%)	19 (8.2%)	
No	283 (88.4%)	71 (79.8%)	212 (91.8%)	
Myocardial infarct				<0.001
Yes	37 (11.6%)	21 (23.6%)	16 (6.9%)	
No	283 (88.4%)	68 (76.4%)	215 (93.1%)	
Respiratory failure				0.155 *
Yes	46 (14.4%)	17 (19.1%)	29 (12.6%)	
No	274 (85.6%)	72 (80.9%)	202 (87.4%)	
Hepatic failure				0.042
Yes	12 (3.8%)	7 (7.9%)	5 (2.2%)	
No	308 (96.3%)	82 (92.1%)	226 (97.8%)	
Hypotension/shock				<0.001
Yes	64 (20.0%)	31 (34.8%)	33 (14.3%)	
No	256 (80.0%)	58 (65.2%)	198 (85.7%)	
Metabolic illness				0.006
Yes	32 (10.0%)	16 (18.0%)	16 (6.9%)	
No	288 (90.0%)	73 (82.0%)	215 (93.1%)	
Diabetes mellitus				< 0.001
Yes	49 (15.3%)	27 (30.3%)	22 (9.5%)	
No	271 (84.7%)	62 (69.7%)	209 (90.5%)	
Pneumonia				0.185 *
Yes	40 (12.5%)	15 (16.9%)	25 (10.8%)	
No	280 (87.5%)	74 (83.1%)	206 (89.2%)	
Sepsis				0.009
Yes	25 97.8%)	13 (14.6%)	12 (5.2%)	
No	295 (92.2%)	76 (85.4%)	219 (94.8%)	
Malignancy				0.131 *
Yes	21 (6.6%)	9 (10.1%)	12 (5.2%)	
No	299 (93.4%)	80 (89.9%)	219 (94.8%)	
Renal failure				0.007
Yes	73 (22.8%)	30 (33.7%)	43 (18.6%)	
No	247 (77.2%)	59 (66.3%)	188 (81.4%)	
Pre-event data				
Cause of cardiac arrest				<0.001
Cardiac	166 (51.7%)	53 (59.6%)	113 (48.7%)	
Respiratory	57 (17.8%)	2 (2.2%)	55 (23.7%)	
Others	98 (30.5%)	34 (38.2%)	64 (27.6%)	
Cardiac arrest process				
Initial rhythm				<0.001
Asystole	59 (18.6%)	13 (14.9%)	46 (20.0%)	
PEA	216 (68.1%)	50 (57.5%)	166 (72.2%)	
VF	21 (6.6%)	13 (14.9%)	8 (3.5%)	
pVT	21 (6.6%)	11 (12.6%)	10 (4.3%)	
Resuscitation time of ROSC				<0.001
0–4 min	56 (17.4%)	32 (36.0%)	24 (10.3%)	
5–9 min	52 (16.2%)	20 (22.5%)	32 (13.8%)	
10–14 min	21 (6.5%)	9 (10.1%)	12 (5.2%)	
15–29 min	50 (15.6%)	18 (20.2%)	32 (13.8%)	
≥30 min	142 (44.2%)	10 (11.2%)	132 (56.9%)	
Postresuscitation process				
Coronary angiography				<0.001
Urgent	22 (8.2%)	15 (16.9%)	7 (3.9%)	
Delayed	15 (5.6%)	12 (13.5%)	3 (1.7%)	
None	231 (86.2%)	62 (69.7%)	169 (94.4%)	
Coronary reperfusion attempted				<0.001
Yes	33 (12.4%)	23 (25.8%)	10 (5.6%)	
No	233 (87.6%)	66 (74.2%)	167 (94.4%)	
Targeted temperature management				0.007
Yes	11 (4.1%)	8 (9.0%)	3 (1.7%)	
No	257 (95.9%)	81 (91.0%)	176 (98.3%)	
CASPRI score				<0.001
0–9	22 (6.9%)	15 (16.9%)	7 (3.0%)	
10–14	55 (17.1%)	26 (29.2%)	29 (12.5%)	
15–19	112 (34.9%)	23 (25.8%)	89 (38.4%)	
≥20	132 (41.1%)	25 (28.1%)	107 (46.1%)	
mean ± SD	18.2 ± 6.0	15.3 ± 6.4	19.4 ± 5.4	<0.001

* *p* < 0.2, included for multivariate analysis. Abbreviations: N.S.: non-significant; OHCA: out-of-hospital cardiac arrest; CPC: Cerebral performance category; PEA: pulseless electrical activity; VF: ventricular fibrillation; pVT: pulseless ventricular tachycardia; ROSC: return of spontaneous circulation; CASPRI: Cardiac arrest survival post-resuscitation in-hospital score.

**Table 2 jcm-10-05131-t002:** Utstein style elements and favorable neurological survival in resuscitated patients.

Patient Data	All (*n* = 168)	Favorable Neurological Survival (*n* = 36)	Others (*n* = 132)	*p* Value
Preexisting conditions				
Myocardial infarct				0.178
Yes	37 (22.0%)	11 (30.6%)	26 (19.7%)	
No	131 (78.0)	25 (69.4%)	106 (80.3%)	
Hypotension/shock				0.033
Yes	65 (38.7%)	8 (22.2%)	57 (43.2%)	
No	103 (61.3%)	28 (77.8%)	75 (56.8%)	
Pre-event data				
Cause of cardiac arrest				0.025
Cardiac	87 (51.8%)	17 (47.2%)	70 (53.2%)	
Respiratory	17 (10.1%)	0 (0.0%)	17 (12.9%)	
Others	64 (38.1%)	19 (52.8%)	45 (34.1%)	
Postresuscitation process				
Coronary angiography				0.004
Urgent	22 (13.1%)	10 (27.8%)	12 (9.1%)	
Delayed	15 (8.9%)	5 (13.9%)	10 (7.6%)	
None	131 (78.0%)	21 (58.3%)	110 (83.3%)	
Coronary reperfusion attempted				0.009
Yes	33 (19.9%)	13 (36.1%)	20 (15.4%)	
No	133 (80.1%)	23 (63.9%)	110 (84.6%)	

**Table 3 jcm-10-05131-t003:** The Cardiac Arrest Survival Post-Resuscitation In-hospital (CASPRI) predictors and favorable neurological survival in resuscitated patients.

Patient Data	CASPRI Points	All (*n* = 168)	Favorable Neurological Survival (*n* = 36)	Others (*n* = 132)	*p* Value
Age group (years)					N.S.
<60	0	63 (37.5%)	13 (36.1%)	50 (37.9%)	
60–69	1	43 (25.6%)	13 (36.1%)	30 (22.7%)	
70–79	2	26 (15.5%)	6 (16.7%)	20 (15.2%)	
≥80	4	36 (21.4%)	4 (11.1%)	32 (24.2%)	
Initial rhythm					<0.001
Witnessed VF/pVT	0	28 (16.7%)	11 (30.6%)	17 (12.9%)	
Non-witnessed VF/pVT	3	8 (4.8%)	5 (13.9%)	3 (2.3%)	
PEA	6	103 (61.3%)	18 (50.0%)	85 (64.4%)	
Asystole	7	29 (17.3%)	2 (5.6%)	27 (20.5%)	
Pre-arrest CPC score					0.039
1	0	64 (38.1%)	19 (52.8%)	45 (34.1%)	
2	2	54 (32.1%)	12 (33.3%)	42 (31.58%)	
3 or 4	9	50 (29.8)	5 (13.9%)	45 (34.1%)	
Resuscitation time of ROSC					N.S.
0–4 min	0	50 (29.8%)	17 (47.2%)	33 (25.0%)	
5–9 min	3	38 (22.6%)	9 (25.0%)	29 (22.0%)	
10–14 min	5	16 (9.5%)	3 (8.3%)	13 (9.8%)	
15–29 min	6	33 (19.6%)	4 (11.1%)	29 (22.0%)	
≥30 min	8	31 (18.5)	3 (8.3%)	28 (21.2%)	
Monitored					N.S.
Yes	0	136 (81.0%)	30 (83.3%)	106 (80.3%)	
No	3	32 (19.0%)	6 (16.7%)	26 (19.7%)	
Factors present prior to arrest					0.004.
None	0	25 (14.9%)	12 (33.3%)	13 (9.8%)	
Renal failure	2	19 (11.3%)	5 (13.9%)	14 (10.6%)	
Mechanical ventilation /Sepsis/Hypotension	3	108 (64.3%)	17 (47.2%)	91 (68.9%)	
Hepatic failure/Malignancy	4	16 (9.5%)	2 (5.6%)	14 (10.6%)	
CAPRI score					<0.001
0–9		21 (12.5%)	11 (30.6)	10 (7.6%)	
10–14		44 (26.2%)	13 (36.1%)	31 (23.5%)	
15–19		51 (30.4%)	8 (22.2%)	43 (32.6%)	
≥20		52 (31.0%)	4 (11.1%)	48 (36.4%)	
mean ± SD		16.5 ± 6.5	11.1 ± 5.6	17.8 ± 6.1	<0.001

Abbreviations: VF: ventricular fibrillation; pVT: pulseless ventricular tachycardia; PEA: pulseless electrical activity; CPC: cerebral performance category; ROSC: return of spontaneous circulation; CASPRI: Cardiac arrest survival post-resuscitation in-hospital score.

**Table 4 jcm-10-05131-t004:** Multivariate logistic regression and AUROCC.

Independent Variable	Odd Ratio	95% Confidence Interval	*p* Value
Survival			
Hypotension/shock	2.3	1.2–4.6	0.014
Metabolic illness	3.4	1.4–7.9	0.005
Resuscitation time of ROSC	0.7	0.6–0.8	<0.001
Coronary angiography	3.6	1.9–6.9	<0.001
Targeted temperature management	8.0	1.8–35.4	0.006
Favorable neurological survival			
Coronary angiography(urgent vs. other groups)	5.5	1.8–16.7	0.003
CASPRI score(0–9 vs. other groups)	9.2	2.2–37.4	0.002
CASPRI score(10–14 vs. other groups)	7.7	2.1–28.2	0.002
AUROCC of CASPRI score	0.77	0.68–0.85	<0.001

Abbreviations: ROSC: return of spontaneous circulation; CASPRI: Cardiac arrest survival post-resuscitation in-hospital score; AUROCC: area under receiver operating characteristic curve.

## Data Availability

Readers could access the data and material supporting the conclusions of the study by contacting Jeffrey Che-Hung Tsai at erdr2181@gmail.com.

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
