# Peer review of "Cardiac Arrest Survival Postresuscitation In-Hospital (CASPRI) Score Predicts Neurological Favorable Survival in Emergency Department Cardiac Arrest"

_jcm, 2021, doi:10.3390/jcm10215131_

Round 1
Reviewer 1 Report
Thank you very much for giving me the opportunity the review the present manuscript, where the authors describe a new scoring system for predicting neurological outcome and survival in IHCA in the emergency department.
The conclusion of this retrospektive analysis is, that a low so called "CASPRI Score" predicts favorable outcome in these patients. Well, due to my opinion it seems relatively clear that a short duration of rescusitation, a younger age, the absense of metabolic disorders and hypotension, all predictors/parameters of the score, influence survival and neurological outcome. Nevertheless, this score may be assistance for medical decisions in the further therapy course a.e. according to the presumed patient-will.
Very interesting is the discussion around the positive influence of TTM, which underlines again the relevance of this tool in the course of treatment in patients after cardiac arrest.
Author Response
Responses to Comments of Reviewer 1:
Thanks for your comments. First, we must emphasize that in our study, “Survival of EDCA was associated with presence of preexisting conditions (heart failure, myocardial infarct, hepatic failure, hypotension/shock, metabolic illnesses, diabetes mellitus, sepsis, and renal failure)” (Line 108), and presence of these medical conditions was an independent predictor of survival after multivariate regression analysis, which may be contrary to common reasonable consideration. We discuss the this finding in Discussions, which is also revised as following paragraph: “Patients with these medical conditions presenting with cardiac arrest in the ED might be in the early stage of time-sensitive deterioration in their disease process, which allowed early recognition and immediate treatment in the ED setting, and leaded to higher chance of survival. While the patients without these medical conditions might lack the reversibility by agile responses in the ED, and had lower chance of survival than their counterpart.”
Second, we appreciate your positive comments on the role of TTM to EDCA. The effectiveness of TTM in patients with IHCA is inconsistent in previous researches, and our study found that its use was favorable for survival in EDCA, but was not predictive of favorable neurological outcome in resuscitated patients.
Reviewer 2 Report
Thank you for the opportunity to review the manuscript by Tsai et al. entitled: “Cardiac Arrest Survival Postresuscitation In-hospital (CASPRI) Score Predicts Neurological Favorable Survival in Emergency Department Cardiac Arrest”. The authors investigated the use of the CASPRI score in a less investigated sub-population of IHCA, emergency department cardiac arrest. The Authors found that that the CASPRI Score was able to detect favourable neurological outcome with an AUROC of 0.77.
The manuscript is well written and organized. I have few comments, they are attached below.
Comments
- Why were following groups excluded? – “pregnant, a prisoner, had a psychiatric disease, or had human immune deficiency virus (HIV) infection” – as reporting from an ED it would be interesting to included literally all patients.
- We used the area under the receiver operating characteristic curve (AUROCC) to com- 86 pare the predictive powers of the scoring systems
- Weren’t OHCA cases excluded? See also table 1. – That is stated in the Material and Methods section.
- What were the causes of CA outside the ED? Where were the patients?
- In the first part of the results it is quite hard to follow which group is reported - can you please provide a flow-chart of the study, maybe with also outcome data?
- Please state p – Values of 0.000 as P < 0.001
- What is the reason that presence of Hypotension/Shock is associated with survival? Generally this is reported as worse sign and associated with worse outcome.
- How was hepatic failure defined?
- Why was TTM only performed in 4%? – This questions the association and inclusion in the regression model
- I did not find other scores in comparison to the CASPRI? This is stated in the methods: “We used the area under the receiver operating characteristic curve (AUROC) to compare the predictive powers of the scoring systems.”
- Can you compare your results to ICU-CA in the discussion (almost same setting) à recently published by Leloup et al (Intensive Care Medicine 2020), Roedl et al (Resuscitation 2020)
- only 52% reached ROSC – were they excluded from the CASPRI analysis and regression analysis?
Author Response
Responses to Comments of Reviewer 2:
1- Why were following groups excluded? – “pregnant, a prisoner, had a psychiatric disease, or had human immune deficiency virus (HIV) infection” – as reporting from an ED it would be interesting to included literally all patients.
Response:
We must admit it was a compromised study design because of our IRB regulation. If we included these patients in our research, the IRB review may no longer be categorized to a simplified review process, and we might be asked to get the patients/family’s informed consents. That might increase greatly the difficulties of our researches. On the other hand, we thought excluding these patients might not influence generalization of the results of our study.
2- We used the area under the receiver operating characteristic curve (AUROCC) to compare the predictive powers of the scoring systems
Response:
Please see response for comments 10.
3- Weren’t OHCA cases excluded? See also table 1. – That is stated in the Material and Methods section.
Response:
We did excluded patients visited ED for OHCA, as described in Materials and Methods, but we included patient who were OHCA in other hospital, but transferred to this hospital after successful resuscitation. The Utstein data definitions for IHCA (reference 10) describe these “OHCA” patients as: ”Did patient have an out-of-hospital arrest leading to this admission.” To clarify this possible misunderstanding, we revise the Material and Methods of our manuscript, and add a sentence “We did not exclude the patients who had been successfully resuscitated from an OHCA event in other hospitals, and were transferred to the ED with spontaneous circulation on arrival“. We also revised the Results as “and 16 (5.0%) had had an initial OHCA event, and was referred to the ED after successful resuscitation in other hospital”.
4- What were the causes of CA outside the ED? Where were the patients?
Response:
Since Taichung Veterans General Hospital is an overcrowding medical center, patients may be transferred to scope rooms and radiology department outside of ED for diagnostic/interventional procedures. Cardiac arrest event may occur when they were receiving gastroduodenoscop, colonoscopy, related interventions, image-guided drainage for abscesses or obstructive lesions, or other invasive procedures.
We revised this sentence as “Six (1.9%) cardiac arrest events occurred outside of the ED when patients were transferred to scope rooms or radiology department for diagnostic/interventional procedures.”
5- In the first part of the results it is quite hard to follow which group is reported - can you please provide a flowchart of the study, maybe with also outcome data?
Response:
Thanks for your comments. We added a flow chart to illustrate the baseline characteristics and outcomes of the patients in Figure 1, and revised the Results: A flowchart of baseline characteristics and outcomes was illustrated in Figure 1.
6- Please state p – Values of 0.000 as P < 0.001
Response:
Thanks for your comments, we have revised all of the expressions of P-value in results and Table (totally 18 changes) according to your opinions.
7- What is the reason that presence of Hypotension/Shock is associated with survival? Generally this is reported as worse sign and associated with worse outcome.
Response:
Thanks for your comments. Our study did find that presence of hypotension/shock, which was one of the worse sign, was associated with survival. We also found that presence of hypotension/shock was also an independent predictor of survival by multivariate logistic regression analysis. We thought that patients with these medical conditions presenting with cardiac arrest in the ED might be more likely to be in the early stage of a disease process, which may allow early recognition and immediate treatment of time-sensitive deterioration of medical conditions in the ED setting (see Discussions). This means that patients without these medical conditions might lack the reversibility by agile response in the ED, so their potential chance of survival was lower than their counterpart. This paragraph is revised as followed: “Patients with these medical conditions presenting with cardiac arrest in the ED might be in the early stage of time-sensitive deterioration in their disease process, which allowed early recognition and immediate treatment in the ED setting, and leaded to higher chance of survival. While the patients without these medical conditions might lack the reversibility by agile responses in the ED, and had lower survival rate than their counterpart.”
8- How was hepatic failure defined?
Response:
Hepatic failure was defined when patients had severe jaundice and high ALT/AST, or had history of liver cirrhosis with ascites, episodes of hepatic encephalopathy or spontaneous peritonitis. We did not specifically described the definitions of these preexisting medical conditions since it will lengthen the manuscript, and they were not routinely specifically defined in previous researches we referred in our manuscript.
9- Why was TTM only performed in 4%? – This questions the association and inclusion in the regression model
Response:
This study was retrospective, and included patients with EDCA from 2014 to 2019. During this study period, TTM had not been widely accepted as routine treatment for suitable patients in our country, so we just truthfully presented the results in our analysis. However, 6% of patients sustaining cardiac arrest receive TTM in Chan’s study (reference 21), and it is 3.7% in Chen’s study (reference 22), so we think the proportion of patients receiving TTM was comparable to other researches. In our study design, variables with a P value of < 0.20 in the univariate analysis were entered into a logistic regression model. For TTM, although it was only performed in 11 patients (4.1%), the P value of chi-squared analysis was 0.07, so it was included in further regression analysis. We found that TTM was an independent predictor for survival in all patients.
10- I did not find other scores in comparison to the CASPRI? This is stated in the methods: “We used the area under the receiver operating characteristic curve (AUROC) to compare the predictive powers of the scoring systems.”
Response:
We actually did not compare the CASPRI score with other scoring system. We revised our description as “We used the area under the receiver operating characteristic curve (AUROCC) to evaluate the predictive powers of the CASPRI scoring system.”
11- Can you compare your results to ICU-CA in the discussion (almost same setting) a recently published by Leloup et al (Intensive Care Medicine 2020), Roedl et al (Resuscitation 2020)
Response:
Thanks for your comment. A paragraph is added in the Discussions to compare the risk factors of survival between EDCA and ICU-CA, as followed.
“Survival from cardiac arrest was higher in EDs than cardiac arrests in intensive care units (ICU-CA), but their risks factors of survival and favorable neurological outcome seem be similar. Treating CA-ICU as unique entity, Roedl and colleagues found that the SOFA score and liver failure after ICU-CA are independent predictors of mortality [25]. Leloup and colleagues found that six-month survival with no or moderate functional sequel are correlated with a number of organ failures ≤ 2 when cardiac arrest occurred, resuscitation time ≤ 5 min, shockable rhythm cardiac arrests, etiology related to the life-sustaining devices in place, absence of preexisting disability or disease deemed fatal within 5 years, and sedation [26]. Our study showed that survival of EDCA was associated with causes of cardiac arrest, shockable rhythm, and short resuscitation time. We also found that post-resuscitation processes (coronary angiography and TTM) were also associated survival, which was not mentioned in researches of ICU-CA.”
12- Only 52% reached ROSC – were they excluded from the CASPRI analysis and regression analysis?
Response:
The data analysis was two folds in our study design. For all patients with EDCA, we evaluated the predictive factors of survival. For patients successfully resuscitated (ROSC at least 20 minutes, 52.2% of all patients), we evaluated which factors could predict favorable neurological survival. The CASPRI scores were evaluated in both of the two steps, and with both of chi-squared and regression analyses. We found that the CASPRI score was associated with survival, and the CASPRI scores were lower in the survival group. We also found the CASPRI scores were lower in the favorable neurological survival group in resuscitated patients.
We included CASPRI score in regression analyses of predictors for survival in all patients and those for favorable neurological outcome in patients successfully resuscitated. The CASPRI score was found to be an independent predictor of favorable neurological outcome in resuscitated patients, but not an independent predictor in analysis for survival in all patients.